# HPV Infection of the Oropharyngeal, Genital and Anal Mucosa and Associated Dysplasia in People Living with HIV

**DOI:** 10.3390/v15051170

**Published:** 2023-05-15

**Authors:** Carmen Hidalgo-Tenorio, Inmaculada Calle-Gómez, Raquel Moya-Megías, Javier Rodríguez-Granges, Mohamed Omar, Javier López Hidalgo, Carmen García-Martínez

**Affiliations:** 1Unit of Infectious Diseases, Hospital Universitario Virgen de las Nieves, Instituto de Investigación Biosanitario de Granada (IBS-Granada), 18014 Granada, Spain; 2Internal Medicine Service, Hospital Universitario Virgen de las Nieves, Instituto de Investigación Biosanitario de Granada (IBS-Granada), 18014 Granada, Spain; 3Microbiology Service, Hospital Universitario Virgen de las Nieves, Instituto de Investigación Biosanitario de Granada (IBS-Granada), 18014 Granada, Spain; 4Unit of Infectious Diseases, Complejo Hospitalario de Jaen, 23007 Jaen, Spain; omarampa@gmail.com; 5Pathology Service, Hospital Universitario Virgen de las Nieves, Instituto de Investigación Biosanitario de Granada (IBS-Granada), 18014 Granada, Spain

**Keywords:** oropharyngeal HPV, cancer, PLHIV, women, MSM

## Abstract

Background: The main objectives were to describe the prevalence of HPV, its genotypes and HPV-associated dysplastic lesions in the oropharyngeal mucosa of PLHIV and related factors. Material and Methods: This cross-sectional prospective study consecutively enrolled PLHIV attending our specialist outpatient units. At visit, HIV-related clinical and analytical variables were gathered, and oropharyngeal mucosa exudates were taken to detect HPV and other STIs by polymerase chain reaction. Samples were also taken from the anal canal of all participants and from the genital mucosa of the women for HPV detection/genotyping and cytological study. Results: The 300 participants had a mean age of 45.1 years; 78.7% were MSM and 21.3% women; 25.3% had a history of AIDS; 99.7% were taking ART; and 27.3% had received an HPV vaccine. HPV infection prevalence in the oropharynx was 13%, with genotype 16 being the most frequent (2.3%), and none had dysplasia. Simultaneous infection with *Treponema pallidum* (HR: 4.02 (95% CI: 1.06–15.24)) and a history of anal HSIL or SCCA (HR: 21.52 (95% CI: 1.59–291.6)) were risk factors for oropharyngeal HPV infection, whereas ART duration (8.8 vs. 7.4 years) was a protective factor (HR: 0.989 (95% CI: 0.98–0.99)). Conclusions: The prevalence of HPV infection and dysplasia was low in the oropharyngeal mucosae. A higher exposure to ART was protective against oral HPV infection.

## 1. Introduction

The survival of people living with human immunodeficiency virus (HIV) (PLHIV) has improved since the introduction of antiretroviral therapy (ART) [1,2,3,4]. HIV infection has become a chronic disease associated with a persistent inflammatory status that increases the probability of complications not related to acquired immunodeficiency syndrome (AIDS), such as non-AIDS defining neoplasms or cardiovascular disease [5,6,7]. Anal squamous cell carcinoma (ASCC) is one of the most frequent non-AIDS-defining neoplasms, with an incidence ranging from 85 × 100,000 p-y in MSM to 22 × 100,000 p-y in women [8]. ASCC is associated with a chronic infection caused by HPV high-risk genotypes, whose prevalence is around 50% in the anal mucosa of women living with HIV (WLHIV) and 74% in men who have sex with men living with HIV (MSMLHIV) [9]. The most frequent AIDS-defining neoplasm in women is HPV-associated cervical carcinoma. Notably, 63.8% of women with this disease in Southern Africa and 27.4% of those in Eastern Africa are HIV-positive, with 9200 and 14,000 new cases per year, respectively [10]. Besides persistent HIV infection, risk factors for the development of cervical, anal and head and neck squamous cell cancers include immunosuppression [11], tobacco [12] and sexual behavior, among others [13]. The risk of oropharyngeal mucosal HPV infection and its complications is greater among PLHIV than in the general population [14,15], but this relationship has been little studied in our country. The study objectives were to determine the prevalence of oropharyngeal mucosal HPV infection, the genotypes involved, infection-related factors and the prevalence of dysplasia in a cohort of PLHIV recruited from participants in a program of anal and cervical lesion screening, diagnosis, treatment and prophylaxis and to compare these data with the prevalence of HPV infection and dysplasia in anal and cervical mucosa.

## 2. Materials and Methods

This prospective cross-sectional study consecutively enrolled 300 persons living with confirmed HIV attended by infectious disease units in the “Hospital Universitario Virgen de las Nieves” (HUVN) and “Hospital Universitario San Cecilio” of Granada and “Complejo Hospitalario” of Jaen and included in a screening program for HPV infection-related anal and cervical dysplasia lesions.

Study inclusion criteria were age ≥18 years and participation in the anal and/or genital cancer screening program of the HUVN. The exclusion criteria included being a pregnant WLHIV.

During the first visit, data were gathered on sex, age and nationality; number of different sexual partners over past 12 months (NP12m); total number of sexual partners since onset of sexual relations (NPT); time in months since the onset of sexual relations; utilization of condoms and percentage utilization (during oral, anal and genital sex); employment situation (active or retired); educational level (illiterate, primary, secondary or university studies); smoker or ex-smoker (packs/year); alcohol consumption (standard drink units (SDUs)); intravenous drug user (IDU) or ex-IDU; HIV acquisition mechanism (MSM, heterosexual or IDU); HPV vaccination status (completed vaccination regimen or not, time since last dose and type of vaccine); months with HIV diagnosis; CDC-classified HIV stage; on-ART or ART-naïve patient; duration of ART (in months); number of ART regimen before first visit; current ART line and months under treatment with this line; virological failure; polypharmacy; presence/history of other infections (chronic liver disease or infection by hepatitis B virus (HBV), chronic liver disease or infection by hepatitis C virus (HCV), and syphilis); HBV vaccination status; presence and history of other sexually transmitted infections (STIs) in anal, female genital and oropharyngeal mucosae; presence and history of condylomas in any localization and their treatment; and history of anal, cervical/vaginal/vulvar and/or oropharyngeal dysplasia and its degree.

Analytical variables considered were nadir CD4+ T cell count, CD4 and CD8 lymphocyte counts and viral load at diagnosis and at baseline. At the same visit, cotton swabs impregnated with physiological saline were used to take anal canal and female genital samples for HPV detection and genotyping using the polymerase chain reaction (PCR) technique (GeneAmp PCR System 9700, Applied Biosystems, Roche, Foster City, CA, USA) and the ThinPrep Pap Test (Thin Prep Processor 2000, Hologic Corp., Marlborough, MA, USA) for cytology. Both samples were analyzed in the hospital pathology laboratory by the same pathologist who validated the HPV PCR results and cytology study. A cotton swab was introduced into the oral cavity and rotated over the internal and external surfaces of the upper and lower lips, the gingival region and the anterior and posterior surfaces of the tongue and the oropharynx; it was then placed in an individual vial with physiological solution. Oropharyngeal mucosa samples were sent to the Microbiology Department for DNA extraction and PCR study of HPV and other STIs (*N. gonorrhoeae*, *Chlamydia trachomatis*, *Trichomonas vaginalis*).

The cytological study used the Bethesda classification to categorize lesions as low-grade (LSIL) or high-grade (HSIL) anal squamous intraepithelial lesions (ASILs), atypical squamous cells of uncertain significance (ASCUS) or uncertain lesions whose high grade cannot be ruled out (ASCUS-H). The Reagan classification was used to categorize cases of ASCUS, LSIL or HSIL as dysplasia, with ASCUS and LSIL being low grade and HSIL high grade [16].

After rectal inspection and digital anorectal examination, all participants underwent high-resolution anoscopy (HRA) with a Carl Zeiss 150 fc © colposcope (Carl Zeiss, Oberkochen, Germany). In brief, 5 mL acetic acid was instilled through a transparent disposable anoscope and left in place for around 3 min before its removal for examination of the mucosa; 5% Lugol’s iodine was then instilled for 1 min before repetition of the anoscopic examination. Samples were selected from quadrants of apparently normal mucosa and from areas with Lugol-negative aceto-white lesions. The biopsies were performed with an endoscopic retrograde cholangiopancreatography (ERCP) catheter.

The histological classification was based on the Lower Anogenital Squamous Terminology (LAST) Standardization Project for HPV, categorizing lesions as LSIL (AIN1/condyloma), HSIL (AIN2, AIN3, C. in situ) or invasive anal squamous cell carcinoma (ASCC) [17].

Genotypes 16, 18, 26, 31, 33, 35, 39, 45, 51–53, 56, 58, 59, 66, 68, 73 and 82 were considered high-risk (HPV-HR) and genotypes 6, 11, 34, 40, 42–44, 54, 55, 57, 61, 70–72, 81, 83, 84 and 89 low-risk (HPV-LR). Genotypes 39, 45, 59 and 68 were classified as subspecies of HPV 18 and genotypes 31, 33, 35, 52, 58 and 67 as subspecies of HPV 16 [18].

Women with abnormal cervical cytology or genital HPV infection were referred to the Gynecology Department for evaluation.

All participants underwent oropharyngeal mucosa inspection before the sampling for HPV PCR, and those with a positive oropharyngeal HPV PCR and/or the presence of symptoms (e.g., chronic voice changes(hoarseness)) and/or visible lesions in the oropharyngeal mucosa were referred to the Otorhinolaryngology Department for examination.

### 2.1. Definition of Variables

The category HSIL-plus includes high-grade lesions (HSILs) and invasive ASCC.

### 2.2. Statistical Analysis

The sample size was calculated using Ene 2.0 statistical software. It was estimated that at least 123 PLHIV were needed to expect a frequency of 25% of oropharyngeal high-risk HPV prevalence in PLHIV, considering previously published global data, with a precision of 5% and a confidence interval of 95% [19].

Descriptive analysis: The means, standard deviations, medians and percentiles were calculated for quantitative variables and the absolute and relative frequencies for qualitative variables. The prevalence of HPV infection was calculated with 95% CI.

A bivariate analysis was conducted on the relationship between possible risk factors for HPV infection and the presence of dysplasia in oropharyngeal mucosa. The Kolmogorov–Smirnov test was used to check the normality of variable distribution. Student’s t-test for independent samples was used for quantitative variables when normally distributed; the Mann–Whitney U test was used when non-normally distributed; and the Wilcoxon test was applied for related quantitative variables. Qualitative variables were analyzed with Pearson’s chi-square test or, when application criteria were not met, Fisher’s test. Finally, a multivariate logistic regression analysis was performed using a stepwise procedure, entering variables found to be significant in bivariate analyses and other factors considered relevant in the literature (sex, age, number of sexual partners throughout life and over previous 12 months, smoking habit, ART experience, STIs in oral mucosa, age at onset of sexual relations, utilization of condom in oral sex, time since HIV diagnosis and receipt of HPV vaccine). SPSS 21.0 (IBM SPSS, Armonk, NY, USA) was used for data analyses, and the level of significance was 0.05 in all tests.

### 2.3. Ethics Approval and Consent to Participate

All participants gave their informed consent to participate in the study, which was approved by the Research Ethics Committee of the hospital (Ref: OROHPV, 0098-N-22). All data were treated in compliance with national data protection legislation (Organic Law 3/2018, 5 December).

## 3. Results

### 3.1. Description of the Cohort

#### 3.1.1. Epidemiological Characteristics

The study included 300 PLHIV (78.7% male) with a mean age of 45.1 years; 87% held Spanish nationality. They had a median of 1 sexual partner over the previous 12 months and a median of 20.5 sexual partners since the onset of sexual relations; 2.7% used condoms during oral sex, 34.3% during vaginal sex and 34.7% during anal sex; and 35.9% of participants were active smokers. HPV vaccination had been received by 31% (tetravalent vaccine by 9% and nonavalent vaccine by 22.7%), and 27.3% had completed the vaccine regimen; 1.7% had chronic HBV infection and 6.3% chronic HCV infection; and 64.3% were vaccinated against HBV. At the first visit (baseline), 5% had active syphilis and 4% anogenital condylomas/warts. Table 1 lists the remaining epidemiological characteristics of the cohort.

#### 3.1.2. Variables Related to HIV Infection

Among participants, 95.6% had acquired HIV by sexual pathway (78.7% were MSM). The median time since HIV diagnosis was 15 years (IQR: 8.5–24.9), 25.3% were in AIDS stage at diagnosis and 99.7% had been receiving ART for a median of 13.5 years (IQR: 7.7–19.5).

Table 2 displays results for the other variables related to HIV infection.

#### 3.1.3. History of Oral, Genital and Anal Mucosa Dysplasia

One MSM-LHIV (0.3%) had had a history of HPV-associated oropharyngeal epidermoid carcinoma five years earlier; 25% of women had a history of genital dysplasia (12.5% CIN1 and 18.8% CIN 2/3); none had a history of cervical squamous carcinoma, while 82% of PLHIV had a history of anal dysplasia (62.3% AIN1, 18% AIN2/3, and 1.7% ASCC), as shown in Table 3.

### 3.2. Current HPV Infection and Oral, Female Genital and Anal Dysplasia

#### 3.2.1. Current HPV Infection and Oral Dysplasia

Among all participants, 39 (13.0%) had HPV infection in oropharyngeal mucosa, 29 (9.7%) with HR serotypes, 15 (5%) with LR serotypes and 5 (1.7%) with simultaneous infection. The most frequently isolated genotypes in patients with oropharyngeal HPV were HR genotypes 16 (2.3%) and 68 (1.7%) and LR genotype 6 (1%). The examination of all participants by the otorhinolaryngologist detected no symptoms or visible lesions in the oropharyngeal area, and no cases of dysplasia were observed in the oropharyngeal mucosa. The presence of STIs other than HPV in the oropharyngeal area was observed in 22 (7.4%) participants: *N. gonorrhoeae* in 20 (90.9%) and *Chlamydia trachomatis* in 2 (9.1). Table 4 lists results for the other variables.

#### 3.2.2. Current HPV Infection and Female Genital Dysplasia

Among the 64 female participants, 31 (50%) had genital HPV infection: 14 (25%) with HR serotypes, 17 (30.4%) with LR serotypes, 7 (12.5%) with HPV coinfection and 3 (5.2%) with simultaneous oropharyngeal and genital HPV infection. The most frequently isolated genotypes in genital mucosa were LR genotypes 62/81 (8.9%) and HR genotypes 16 and 42 (7.2% for both).

Cervical cytology was normal in 55 women (88.7%), ASCUS in 4 (1.3%) and LSIL in 3 (1.0%). Gynecological evaluation detected CIN1 in one woman (0.7%). Appendix A lists results for the other variables.

#### 3.2.3. Current HPV Infection and Anal Dysplasia

Among the 300 PLHIV, 238 (82.6%) had HPV infection of the anal mucosa: 189 (67.3%) with HR serotypes, 168 (59.8%) LR serotypes and 122 (43.8%) with coinfection. Simultaneous HPV infection in oropharyngeal and anal mucosa was observed in 25 (8.9%). The most frequent genotypes in anal mucosa were LR 44/55 and 62/81 (prevalence of 19.4% for both) and HR 16 and 68 (14.7% for both).

Anal cytology was normal in 118 participants (39.6%), LSIL in 110 (36.9%), HSIL in 8 (2.7%) and ASCUS in 33 (11.1%). HRA and biopsy results revealed LSIL (AIN1) in 84 (29.2%) (19 (22.3%) in women and 65 (77.4%) HSH) and HSIL plus (AIN2/AIN3/Carcinoma in situ) in 5 (1.7%) (1 (20%) in women and 4 (80%) MSM). Appendix A displays the results for the other variables.

### 3.3. Factors Related to HPV Infection in Oropharyngeal Mucosa

In bivariate analyses, simultaneous infection by *Treponema pallidum* (12.8 vs. 3.8%, *p* = 0.03) and a history of anal HSIL-plus (7.7 vs. 0.8%; *p* = 0.017) were associated with oropharyngeal HPV infection. In the multivariate analysis, simultaneous infection by *Treponema pallidum* (HR 4.02 (95% CI: 1.06–15.24); *p* = 0.04) and a history of HSIL-plus (HR 21.52; (95% CI: 1.59–291.6); *p* = 0.02) emerged as risk factors, while a longer duration of ART (8.8 vs. 7.4 years (HR: 0.989 (95% CI: 0.98–0.99)); *p* = 0.034) emerged as a protective factor. Table 5 lists results for the remaining variables.

## 4. Discussion

HPV infection of the oropharyngeal mucosa was infrequent in this cohort of PLHIV and less prevalent than anal or cervical HPV infections. Genotype 16 was the most frequently isolated genotype (2.3%), habitually as a monoinfection with no associated dysplasia. The most frequent genotypes in female genital mucosa were LR, including genotype 44/55, while the most frequent in the anus were HR-HPV, with a predominance of genotypes 16 and 68. A South American study of eight PLHIV with HPV-associated oropharyngeal lesions found that patients had multiple HPV infections, largely involving HR genotypes 16, 52 and 56 [20]. In a European study, 69% of a series of MSM living with HIV had at least one of the seven HR genotypes studied (16, 18, 31, 33, 45, 52, 58) in anal, oral and penile mucosae [21].

Among the mucosae examined in the present investigation, the highest prevalence and degree of dysplasia was observed in anal mucosa samples, with almost 30% diagnosed by HRA with LSIL/AIN1 and 1.7% with HSIL. No participant presented oropharyngeal dysplasia, and only 1.7% of the women had CIN1. The risk of HPV-related anal cancer is higher in PLHIV than in the general population [22]. Its greater frequency in comparison to oropharyngeal–laryngeal squamous cancer may be attributable to the higher incidence and longer persistence of this infection in the anal canal [23]. This is possibly due to the protective effect of metalloproteinases (MMP-8) in saliva against HPV infection [24] In contrast, the mucosa of the anus does not present any lubricating or protective substance. The common risk factors and acquisition mechanisms of oropharyngeal HPV infection were a lifetime history of more than six sex partners, tobacco consumption, age >50 and oral sex, among others [25].

In the present study, the prevalence of HPV was up to 6-fold higher in anal versus oral mucosae. A recent study of 103 MSM-LHIV described a prevalence of oral HPV infection of 14% and no associated dysplastic lesions, similar to the present findings, with an anal HPV infection prevalence of 88.3% and an HSIL prevalence of 24.3% [26].

In this cohort of PLHIV, concomitant infection by *Treponema pallidum* and a history of HSIL-plus in the anal canal emerged as risk factors for oropharyngeal HPV infection. Simultaneous infection with HPV and other STIs is frequent in PLHIV because they share common risk factors and acquisition mechanisms [27]. Studies in the general population have associated the prevalence of oropharyngeal mucosal HPV with male sex, older age, more sexual partners since starting sexual relations and an active smoking habit [28]. Another study of 170 MSM (including 72 PLHIV) found risk factors for oral HPV infection to be the commencement of oral sexual relations at the age of 18 years and a lifetime experience of more than 50 receptive oral sex partners [29].

A longer time with ART proved to be a protective factor against oropharyngeal HPV infection in the present study, in line with previous findings of a lower oral mucosal infection rate after 12 months of ART [30]. This is consistent with reports associating HPV infection in oropharyngeal [23], anal [31] and cervical [32] mucosae of PLHIV with low nadir CD4 [31] and current CD4+ T cell count [32]. Hence, ART appears to protect against HPV infection at all of these mucosal sites.

The main limitations of this study are derived from its design and the specific population, hampering the extrapolation of data to other types of populations. Its strengths include the largest sample size to date for this type of study, the methodology and systematic analysis applied and its novelty: it is one of the few prospective investigations of the characteristics of HPV infection in the oropharyngeal mucosa of PLHIV.

## 5. Conclusions

HPV infection has a low prevalence in the oropharyngeal mucosa of PLHIV in our setting, with a predominance of high-risk genotypes; it is not associated with dysplasia. Simultaneous infection by *Treponema pallidum* and a history of anal cancer precursor lesions or ASCC emerged as risk factors for this infection, while greater exposure to ART was a protective factor.

## Figures and Tables

**Table 1 viruses-15-01170-t001:** Epidemiological characteristics of the cohort.

	*n* = 300
Age, mean (years), (±SD)	45.1 (11)
Sex, *n* (%)	
Males (MSM)	236 (78.7)
Females	64 (21.3)
Spanish nationality, *n* (%)	261 (87)
NSP12m median (IQR)	1 (1–3)
TNSP median (IQR)	20.5 (4–298)
Median of months since onset of SR (IQR)	390 (228–474)
Use of condom for oral sex, *n* (%)	8 (2.7)
Use of condom for vaginal sex, *n* (%)	12 (34.3)
Use of condom for anal sex, *n* (%)	104 (34.7)
Retired, *n* (%)	35 (11.7)
Educational level, *n* (%)	
-Illiterate	12 (4.0)
-Primary	67 (22.3)
-Secondary	92 (30.7)
-University	129 (43.0)
Smokers, *n* (%)	107 (35.9)
Ex-smokers, *n* (%)	72 (24.1)
Median packs/year, (IQR)	3.13 (0–18.8)
Alcohol, *n* (%)	35 (11.7)
Median SDUs, (IQR)	0 (0–0)
IDU, *n* (%)	2 (0.7)
Ex-IDU, *n* (%)	18 (6.0)
Polypharmacy, *n* (%)	25 (8.3)
HPV vaccine, *n* (%)	93 (31.0)
-Complete vaccination	82 (27.3)
-Tetravalent vaccine	27 (9.0)
-Nonavalent vaccine	68 (22.7)
Chronic HBV infection, *n* (%)	5 (1.7)
Active chronic HCV infection, *n* (%)	19 (6.3)
Syphilis at baseline, *n* (%)	15 (5.0)
History of syphilis, *n* (%)	124 (41.3)
Other STIs at baseline, *n* (%)	27 (9.0)
History of other STIs, *n* (%)	102 (34.0)
Condylomas at baseline, *n* (%)	12 (4.0)
History of genital condylomas, *n* (%)	93 (31.0)
History of condyloma removal, *n* (%)	
-Imiquimod	36 (43.4)
-Surgery	14 (16.9)
-Cryotherapy	11 (13.3)
History of condyloma relapse, *n* (%)	25 (29.1)

IQR, interquartile range; NSP12m, number of sexual partners over past 12 months; TNSP, total number of sexual partners since onset of sexual relations; SR, sexual relations; SDUs, standard drink units; IDU, intravenous drug user; HPV, human papillomavirus; HBV, hepatitis B virus; HCV, hepatitis C virus; STI, sexually transmitted infection; TBC, tuberculosis; LTBI, latent tuberculosis infection.

**Table 2 viruses-15-01170-t002:** Variables related to HIV infection.

	*n* = 300
HIV infection acquisition mechanism, *n* (%)	
-MSM	233 (78.7)
-Heterosexual	50 (16.9)
-IDU	12 (4.1)
-Unknown	5 (1.7)
Median years since HIV diagnosis (IQR)	15 (8.5–24.9)
AIDS (A3, B3, C)	74 (25.3)
CD4 at diagnosis, mean (±SD)	398.9 (±276.7)
CD8 at diagnosis, mean (±SD)	1.006.8 (±492.6)
VL at diagnosis, mean (±SD)	5.46 (±6.03)
Current CD4, mean (±SD)	745.6 (±328)
Current CD8, mean (±SD)	900.5 (±467.2)
Current CD4/CD8, mean (±SD)	0.97 (±0.6)
Current undetectable VL, *n* (%)	239 (82.4)
Naïve, *n* (%)	1 (0.3)
Median of years with ART since onset (IQR)	13.5 (7.7–19.5)
Median of ART regimens since onset (IQR)	4 (3–6)
Median of months with current ART (IQR)	26 (12–40.5)
Current ART, *n* (%)	299 (99.7)
Virological failure, *n* (%)	3 (1.0)

IQR, interquartile range; HIV, human immunodeficiency virus; AIDS, acquired immunodeficiency syndrome; MSM, men who have sex with men; IDU: intravenous drug user; CD4, CD4 lymphocytes; CD8, CD8 lymphocytes; VL, viral load; CD4/CD8, CD4/CD8 ratio; ART, antiretroviral therapy; PI, protease inhibitor; II, integrase inhibitor.

**Table 3 viruses-15-01170-t003:** History of oral, genital and anal mucosa dysplasia.

	*n* = 300
History of oral dysplasia, *n* (%)	
-Epidermoid carcinoma	1 (0.3)
History of genital dysplasia, *n* (%)	
-CIN 1	8 (12.5)
-CIN2–3	12 (18.8)
-Cervical cancer	0 (0)
History of anal dysplasia, *n* (%)	
-LSIL (AIN 1)	187 (62.3)
-HSIL (AIN 2–3)	54 (18)
-Anal cancer	5 (1.7)

HPV, human papillomavirus; CIN, cervical intraepithelial neoplasm; AIN, anal intraepithelial neoplasm.

**Table 4 viruses-15-01170-t004:** Current HPV infection and the presence of other STIs and oropharyngeal dysplasia.

	*n* = 300
HPV PCR in oropharyngeal mucosa, *n* (%)	39 (13.0)
HPV-HR, *n* (%)	29 (9.7)
Number of HPV-HR serotypes, median (IQR)	0 (0–0)
HPV-LR, *n* (%)	15 (5.0)
Number of HPV-LR serotypes, median (IQR)	0 (0–0)
HPV coinfection, *n* (%)	5 (1.7)
HPV 1, *n* (%)	1 (0.3)
HPV 6, *n* (%)	3 (1.0)
HPV 11, *n* (%)	1 (0.3)
HPV 16, *n* (%)	7 (2.3)
HPV 18, *n* (%)	2 (0.7)
HPV 26, *n* (%)	1 (0.3)
HPV 31, *n* (%)	1 (0.3)
HPV 33, *n* (%)	2 (0.7)
HPV 35, *n* (%)	1 (0.3)
HPV 39, *n* (%)	4 (1.3)
HPV 40, *n* (%)	3 (1.0)
HPV 42, *n* (%)	2 (0.7)
HPV 43, *n* (%)	1 (0.3)
HPV 44, *n* (%)	4 (1.3)
HPV 51, *n* (%)	1 (0.3)
HPV 52, *n* (%)	1 (0.3)
HPV 53, *n* (%)	1 (0.3)
HPV 54, *n* (%)	2 (0.7)
HPV 56, *n* (%)	2 (0.7)
HPV 58, *n* (%)	1 (0.3)
HPV 59, *n* (%)	3 (1.0)
HPV 61, *n* (%)	2 (0.7)
HPV 62/81, *n* (%)	1 (0.3)
HPV 66, *n* (%)	2 (0.7)
HPV 68, *n* (%)	5 (1.7)
HPV 69, *n* (%)	2 (0.7)
HPV 70, *n* (%)	2 (0.7)
HPV 82, *n* (%)	1 (0.3)
Oral HPV-related dysplasia, *n* (%)	0
Positive PCR for other STIs, *n* (%)	22 (7.4)
*-N. gonorrhoeae*	20 (90.9)
*-C. trachomatis*	2 (9.1)

IQR, interquartile range; PCR, polymerase chain reaction; HPV, human papillomavirus; STI, sexually transmitted infection.

**Table 5 viruses-15-01170-t005:** Risk factors for oropharyngeal HPV. Bivariate and multivariate analyses.

	Positive Oral HPV *n* = 39	Negative Oral HPV *n* = 261	*p*	HR (95% CI) *p* *
Age, mean (±SD)	45.4 (11.5)	45 (10.9)	0.87	0.96 (0.85–1.07); 0.45
45 (10.9)				
Male	31 (79.5)	205 (78.5)	0.89	
Female	8 (20.5)	56 (21.5)	0.89	0.94 (0.24–3.61); 0.93
Employment situation, *n* (%)				
Active	36 (92.3)	229 (87.7)	0.59	
Retired	3 (7.7)	32 (12.3)	0.59	
Unemployed	0 (0)	0 (0)	0.59	
Educational level, *n* (%)				
Illiterate	2 (5.1)	10 (3.8)		
Primary	10 (25.6)	57 (21.8)		
Secondary	11 (28.2)	81 (31.0)	0.92	
University	16 (41.0)	113 (43.3)		
Spanish nationality, *n* (%)	32 (82.1)	229 (87.7)	0.42	
Median NSP12m (IQR)	1 (0–4)	1 (1–5)	0.51	0.99 (0.99–1.01); 0.7
Median TNSP (IQR)	150 (14–357)	50 (16–200)	0.17	1 (1–1); 0.77
Median (months) onset of SR (IQR)	348 (264–440)	312 (216–420)	0.27	1.01 (0.99–1.02); 0.29
Use of condom oral sex, *n* (%)	1 (2.6)	7 (2.7)	1.00	0.91 (0.1–8.63); 0.93
Use of condom vaginal sex, (%)	2 (50.0)	10 (32.3)	0.59	
Use of condom anal sex, (%)	13 (33.3)	91 (34.9)	0.85	
Smokers, *n* (%)	16 (41.0)	91 (35.1)	0.48	1.41 (0.63–3.18); 0.41
Ex-smokers, *n* (%)	8 (20.5)	64 (24.6)	0.57	
Median packs/year (IQR)	3.25 (0–7)	6.5 (0–19.5)	0.74	
Alcohol, *n* (%)	7 (17.9)	28 (10.8)	0.19	
Median SDUs (IQR)	0 (0–0)	0 (0–0)	0.34	
IDU, *n* (%)	0 (0)	2 (0.8)	1.00	
Ex-IDU, *n* (%)	3 (7.9)	15 (5.8)	0.71	
Acquisition of HIV, *n* (%)				
-MSM	31 (79.5)	202 (78.6)		
-Heterosexual	6 (15.4)	44 (17.1)	0.95	
-IDU	2 (5.1)	10 (3.9)		
History of syphilis, *n* (%)	20 (51.3)	104 (40.0)	0.18	
History of other STIs, *n* (%)	17 (43.6)	83 (31.9)	0.15	
History of genital condylomas, *n* (%)	14 (35.9)	79 (30.3)	0.48	
History of oral dysplasia, *n* (%)	0	0		
History of anal dysplasia, *n* (%)	25 (64.1)	168 (64.4)	0.97	
-AIN1	11 (28.9)	73 (29.2)	0.98	
-HSIL-plus	3 (7.7)	2 (0.8)	0.017	21.52 (1.59–291.6); 0.02
History of female genital dysplasia	2 (25%)	14 (25%)	1	
Chronic HBV infection, *n* (%)	0	5 (1.9)	1	
Active chronic HCV infection, *n* (%)	3 (7.7)	16 (6.1)	0.72	
Cured HCV infection, *n* (%)	5 (12.8)	22 (8.4)	0.37	
HPV vaccination, *n* (%)				
-Vaccinated	10 (25.6)	83 (31.8)	0.44	1.16 (0.38–3.49); 0.79
-Complete vaccination	7 (17.9)	75 (28.7)	0.16	
-Tetravalent vaccination	1 (2.6)	26 (10.0)	0.23	
-Nonavalent vaccination	10 (25.6)	58 (22.2)	0.63	
Median HIV (months) (IQR)	130 (45–186.5)	128 (72–219)	0.68	1.01 (0.99–1.02); 0.091
AIDS (stage A3, B3, C), *n* (%)	8 (21.6)	66 (25.9)	0.58	
CD4 at HIV diagnosis, mean (±SD)	399.2 (271.9)	398.8 (278.1)	0.99	
CD8 at HIV diagnosis, mean (±SD)	840.5 (351.7)	1,030 (505.8)	0.11	
CD4 nadir, mean (±SD)	349.5 (267.1)	327.0 (244.9)	0.59	
Current CD4, mean (±SD)	709.4 (302.1)	751.0 (331.9)	0.47	
Current CD8, mean (±SD)	973.3 (339.6)	889.4 (476.4)	0.30	
Current undetectable VL, *n* (%)	29 (76.3)	210 (83.3)	0.29	
Naïve, *n* (%)	0 (0)	1 (0.4)	1.00	
Median (months) with ART (IQR)	88.5 (29–164)	106 (63–174.5)	0.16	0.989 (0.98–0.99); 0.034
Median ART lines (IQR)	3 (2–5)	3 (2–5)	0.94	
Virological failure, *n* (%)	1 (2.6)	2 (0.8)	0.34	
Polypharmacy, *n* (%)	4 (10.3)	21 (8.0)	0.55	
Syphilis at baseline, *n* (%)	5 (12.8)	10 (3.8)	0.03	4.02 (1.06–15.24); 0.04
STI in oral mucosa at baseline, *n* (%)	5 (13.2)	17 (6.5)	0.18	2.14 (0.61–7.54); 0.24
Condylomas at baseline, *n* (%)	2 (5.1)	10 (3.8)	0.66	
Positive genital HPV PCR, *n* (%)	3 (42.9)	28 (51.9)	0.71	
Positive anal HPV PCR, *n* (%)	33 (89.2)	205 (82.0)	0.28	

IQR, interquartile range; NSP12m, number of sexual partners over past 12 months; TNSP, total number of sexual partners since onset of sexual relations; SRs, sexual relations; SDUs, standard drink units; IDU: intravenous drug user; HIV, human immunodeficiency virus; MSM, men who have sex with men; HPV, human papillomavirus; AIDS, acquired immunodeficiency syndrome; CD4, CD4 lymphocytes; CD8, CD8 lymphocytes; VL, viral load; CD4/CD8, CD4/CD8 ratio; ART, antiretroviral therapy; HBV, hepatitis B virus; HCV, hepatitis C virus; STI, sexually transmitted infection; PCR, polymerase chain reaction. *p* *: *p-Value* <0.005

## Data Availability

The authors declare that the database will be made available on request.

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
