# Peer review of "HPV Infection of the Oropharyngeal, Genital and Anal Mucosa and Associated Dysplasia in People Living with HIV"

_viruses, 2023, doi:10.3390/v15051170_

Round 1

Reviewer 1 Report

The study describes HPV prevalence and genotypes and the prevalence of HPV-associated dysplastic lesions in the oropharyngeal mucosa of PLHIV and related factors. The article provides good information on the topic and authors have taken appreciable efforts on multiple aspects of this study.

General comments: 

1.     Under Introduction please include recent and global epidemiological data on HIV-HPV coinfection and percentage of ASCC or cervical cancer in HIV infected men and women at global level. 

2.     It would be better to include a brief note on HIV pathology and factors attributed to development of cancers in genital, anal and oropharyngeal area and etiological agents responsible for cancer in Oropharyngeal Mucosa, highlighting the rationale behind this study. 

3.     For better understanding of study design and partipants: Add inclusion and exclusion criteria for study participants with a clear mention on participation/ exclusion of pregnant women in this study.

4.     For easy and quick understanding of data try pie chart representation wherever appropriate.   

Few minor points that need to be addressed are mentioned below: 

Please check for the terminologies and formation of abbreviations [e.g. Virgen de las Nieves University Hospital, abbreviated in HUVN] used in the manuscript. Use all terminologies in the same chronological order for better and faster understanding of data.

Line no 52: Please mention 300 patients with HIV as “Confirmed HIV positive cases/patients”  

Line no 60:  Does the statement ‘total number of sexual partners since onset of sexual relations (NPT)’, mean total number of sexual partners since last intercourse?  

Line no 67-68: It is more appropriate to use word ‘HIV diagnosis’ rather than ‘AIDS diagnosis’ because AIDS is the last stage of HIV infection. Also, use the term as on-ART patient and ART-naïve patient.  Months under ART shall be Duration of ART (in months) and please change ART line as ART regimen. 

Line no 77: The terms viral load at diagnosis and at baseline means the same. 

Line no 79-80: It would be great to include name and manufacturer of the PCR and Genotyping kits used. Also mention the details of HPV-PCR like standard or multiplex, genes targeted, primers, PCR conditions etc.; along with appropriate citation. 

In Line no 83: Include the sample collection method of Oropharyngeal mucosa. Briefly mention the reason behind selecting these samples for diagnosing STIs.

In lines 86-90: It would be great to briefly mention on cytology studies. It must be a PAP staining.  Please discuss dysplasia and its relationship with cancer in more detail .To my understanding dysplasia denotes the presence of abnormal cells within a tissue. However, it doesn’t necessarily denote cancer. It may lead to cancer, sometimes. 

Line No 110-114: Patients with HPV-PCR positive report or showing symptoms of change in voice were sent to Otorhinolaryngology Department. Please mention the rationale behind this consultation and significance of ‘change in voice’ along with HSIL and LSIL (whether they indicate chronic and acute HPV infection) in the context of HPV infection. 

Table 1 shows that 31% patients were HPV-vaccinated. If possible represent data correlating these patients with clinical examination, cytological and molecular diagnosis. 

In Table 2 please check the total of HIV infection acquisition mechanism. MSM (233) +Heterosexual (50) +IDU (12) = N300, but the table shows 5 less. 

Table 3 carries missing Data in last two rows HSIL (AIN 2-3) and Anal cancer. 

In all three tables: please explain what the bracket denotes.

Please verify the most common HPV genotypes on line no 198-199, by matching HR and LR grouping with line no 104. 

In line no 210 there is a mention on biopsy results but the corresponding information is missing from Materials and Methods section. 

Line no. 255:  It would be good to discuss in brief the common risk factors and acquisition mechanisms.

Acceptable

Author Response

Responses to Reviewer 1

The study describes HPV prevalence and genotypes and the prevalence of HPV-associated dysplastic lesions in the oropharyngeal mucosa of PLHIV and related factors. The article provides good information on the topic and authors have taken appreciable efforts on multiple aspects of this study.

We are very grateful to this reviewer for these comments, which have helped us to strengthen our article.  

General comments: 

  1. Under Introduction please include recent and global epidemiological data on HIV-HPV coinfection and percentage of ASCC or cervical cancer in HIV infected men and women at global level. 
  2. It would be better to include a brief note on HIV pathology and factors attributed to development of cancers in genital, anal and oropharyngeal area and etiological agents responsible for cancer in Oropharyngeal Mucosa, highlighting the rationale behind this study. 

Points 1 and 2 are addressed in the following paragraph, which has been modified accordingly:

Anal squamous cell carcinoma (ASCC) is one of the most frequent non-AIDS-defining neoplasms, with an incidence ranging from 85 x 100,000 p-y in MSM to 22 x 100,000p-y in women. (8-Clifford G, George D, Shiels MS, Engels E A, Albuquerque A, Pynten IM et al et al A meta-analysis of anal cancer incidence by risk group: Toward a unified anal cancer risk scale Int J Cancer. 2021; 148: 38-47).

ASCC is associated with chronic infection caused by HPV high-risk genotypes, whose prevalence is around 50% in the anal mucosa of women living with HIV (WLHIV) and 74% in men who have sex with men living with HIV (MSMLHIV) (9-Wei F, Gaisa MM, D'Souza G, Xia N, Giuliano AR, Hawes SE, et al. Epidemiology of anal human papillomavirus infection and high-grade squamous intraepithelial lesions in 29 900 men according to HIV status, sexuality, and age: a collaborative pooled analysis of 64 studies. Lancet HIV 2021; 8: e531-e543).

The most frequent AIDS-defining neoplasm in women is HPV-associated cervical carcinoma. Notably, 63.8% of women with this disease in Southern Africa and 27.4% of those in Eastern Africa are HIV-positive, with 9,200 and 14,000 new cases per year, respectively (10-Stelzle D, Tanaka LF, Lee KK, Ibrahim Khalil A, Baussano I, Shah ASV, et al. Estimates of the global burden of cervical cancer associated with HIV. Lancet Glob Health 2021; 9: e161-e169).

Besides persistent HIV infection, risk factors for the development of cervical, anal, and head and neck squamous cell cancers include immunosuppression (11- Eng C, Ciombor KK, Cho M, Dorth JA, Rajdev LN, Horowitz DP et al. Anal Cancer: Emerging Standards in a Rare Disease. J. Clin Oncol 2022; 40: 2774-2788), tobacco (12-Umutoni V, Schabath MB, Nyitray AG, Wilkin TJ, Villa LL, Lazcano-Ponce E, et al. The Association between Smoking and Anal Human Papillomavirus in the HPV Infection in Men Study. Cancer Epidemiol Biomarkers Prev. 2022 Aug 2;31(8):1546-1553), and sexual behavior, among others (13-Colón-López V, Shiels MS, Machin M, Ortiz AP, Strickler H, Castle PE, et al. Anal Cancer Risk Among People With HIV Infection in the United States. J Clin Oncol.2018; 36: 68-75).

The risk of oropharyngeal mucosal HPV infection and its complications is greater among PLHIV than among the general population (14,15), but this relationship has been little studied in our country. The study objectives were to determine the prevalence of oropharyngeal mucosal HPV infection, the genotypes involved, infection-related factors, and the prevalence of dysplasia in a cohort of PLHIV recruited from among participants in a program of anal and cervical lesion screening, diagnosis, treatment, and prophylaxis, and to compare these data with the prevalence of HPV infection and dysplasia in anal and cervical mucosa.

  1. For better understanding of study design and participants: Add inclusion and exclusion criteria for study participants with a clear mention on participation/ exclusion of pregnant women in this study.

The following inclusion and exclusion criteria have been included in Material and Methods:

This prospective cross-sectional study consecutively enrolled 300 persons living with HIV attended by infectious disease units in the “Hospital Universitario Virgen de las Nieves” (HUVN) and “Hospital Universitario San Cecilio” of Granada and the “Complejo Hospitalario” of Jaen and included in a screening program for HPV infection-related anal and cervical dysplasia lesions.

Study inclusion criteria were age 18 years and participation in the HUVN anal and/or genital cancer screening program. Pregnant WLHIV were excluded from this study.

  1. For easy and quick understanding of data try pie chart representation wherever appropriate.   

We have carefully considered this proposal but do not consider pie chart representation to be necessary in this paper.

Few minor points that need to be addressed are mentioned below: 

Please check for the terminologies and formation of abbreviations [e.g. Virgen de las Nieves University Hospital, abbreviated in HUVN] used in the manuscript. Use all terminologies in the same chronological order for better and faster understanding of data. These changes have been made.

Line no 52: Please mention 300 patients with HIV as “Confirmed HIV positive cases/patients.” We have changed this to: 300 persons living with confirmed HIV

Line no 60:  Does the statement ‘total number of sexual partners since onset of sexual relations (NPT)’, mean total number of sexual partners since last intercourse?  

It refers to the number of people with whom PLHIV have had sex in their lifetime.

Line no 67-68: It is more appropriate to use word ‘HIV diagnosis’ rather than ‘AIDS diagnosis’ because AIDS is the last stage of HIV infection. Also, use the term as on-ART patient and ART-naïve patient.  Months under ART shall be Duration of ART (in months) and please change ART line as ART regimen. 

These changes have all been made.

Line no 77: The terms viral load at diagnosis and at baseline means the same. 

This is not the case in this paper. We have clarified that “baseline” refers to the time point when participants were enrolled in this study.

Line no 79-80: It would be great to include name and manufacturer of the PCR and Genotyping kits used. Also mention the details of HPV-PCR like standard or multiplex, genes targeted, primers, PCR conditions etc.; along with appropriate citation. 

This has been done, as follows: (lines 83-87): At the same visit, cotton swabs impregnated with physiological saline were used to take anal canal and female genital samples for HPV detection and genotyping by polymerase chain reaction (PCR) technique (GeneAmp PCR System 9700, Applied Biosystems, Roche), and for cytology using the ThinPrep Pap Test (Thin Prep Processor 2000, Hologic Corp).

In Line no 83: Include the sample collection method of Oropharyngeal mucosa. Briefly mention the reason behind selecting these samples for diagnosing STIs.

It is now explained that this allowed the microbiology laboratory to obtain two PCR determinations (for STIs and HPV) from a single sample.

In lines 86-90: It would be great to briefly mention on cytology studies. It must be a PAP staining.  Please discuss dysplasia and its relationship with cancer in more detail .To my understanding dysplasia denotes the presence of abnormal cells within a tissue. However, it doesn’t necessarily denote cancer. It may lead to cancer, sometimes. 

It is reported in Material and Methods that we used the Bethesda classification for the cytology study (Lines 95-97).

Line No 110-114: Patients with HPV-PCR positive report or showing symptoms of change in voice were sent to Otorhinolaryngology Department. Please mention the rationale behind this consultation and significance of ‘change in voice’ along with HSIL and LSIL (whether they indicate chronic and acute HPV infection) in the context of HPV infection. 

This part now reads: Patients with HPV infection in the oropharyngeal mucosa, chronic voice changes (hoarseness), and/or visible lesions in the oropharyngeal mucosa were referred to an otolaryngologist for examination.

Table 1 shows that 31% patients were HPV-vaccinated. If possible represent data correlating these patients with clinical examination, cytological and molecular diagnosis. 

This is an interesting proposal. However, it would make the article longer and was not a study objective. In this regard, our bivariate and multivariate analyses of the risk factors for oropharyngeal HPV infection, reported in Table 5, showed that the HPV vaccine did not protect against oral HPV infection in this cohort of PLHIV.

In Table 2 please check the total of HIV infection acquisition mechanism. MSM (233) +Heterosexual (50) +IDU (12) = N300, but the table shows 5 less. 

This is because the mechanism of acquisition is unknown in these five cases, as now clarified in the revised table.

Table 3 carries missing Data in last two rows HSIL (AIN 2-3) and Anal cancer. 

In all three tables: please explain what the bracket denotes.

This was an error in the configuration of the tables. There were no missing data.

Please verify the most common HPV genotypes on line no 198-199, by matching HR and LR grouping with line no 104. 

This has been done.

In line no 210 there is a mention on biopsy results but the corresponding information is missing from Materials and Methods section. 

This information is given in the following section:

2.1. Definition of variables: The category HSIL-plus includes high-grade lesions (HSILs) and invasive ASCC, (Lines 123-124)

Line no. 255:  It would be good to discuss in brief the common risk factors and acquisition mechanisms.

This issue is briefly discussed in lines 251-253: Its greater frequency in comparison to oropharyngeal-laryngeal squamous cancer may be attributable to the higher incidence and longer persistence of this infection in the anal canal (23).

Reviewer 2 Report

This is a very useful study even though many of these findings have been previously reported. This study is more thorough than previous ones.

1.  Line 27. HSIL should be anal HSIL.

2. "in our setting"...meaning what exactly? (line 44)

3. Line 92. Should be digital anorectal.

4. "Endoscopic" ?? Explain. Line 98

5. Line 83. Explain exactly how the oral specimen was taken. Important. Mouth washing may be better than swab.

6. HRA and Pap are standard and need not be in such detail.

7.  Table 1. A lot of non-relevant items.

8. Table 3. Numbers are wrong for the anal pathology.

9. The title relates to ORAL but the article is on GENITAL and ORAL.

Author Response

RESPONSES TO REVIEWER 2 

This is a very useful study even though many of these findings have been previously reported. This study is more thorough than previous ones.

We are grateful for these positive remarks and for the very helpful suggestions, as detailed below.

  1.  Line 27. HSIL should be anal HSIL. This change has been made.
  2. "in our setting"...meaning what exactly? (line 44). “Setting” has been changed to “country”.
  3. Line 92. Should be digital anorectal. This change has been made.
  4. "Endoscopic" ?? Explain. Line 98. This has been rewritten as follows: “Biopsies were performed with an endoscopic retrograde cholangiopancreatography (ERCP) catheter“
  5. Line 83. Explain exactly how the oral specimen was taken. Important. Mouth washing may be better than swab. The procedure is now fully described, as follows:

A cotton swab was introduced into the oral cavity and rotated over the internal and external surfaces of the upper and lower lips, the gingival region, the anterior and posterior surfaces of the tongue, and the oropharynx; it was then placed in an individual vial with physiological solution for DNA extraction and detection of HPV by PCR.

  1. HRA and Pap are standard and need not be in such detail. If acceptable to the editorial team, we would prefer to retain this information in order to assist readers who might not be familiar with these concepts.
  2.  Table 1. A lot of non-relevant items. We have simplified the table by reducing the number of items.
  3. Table 3. Numbers are wrong for the anal pathology. Table 3 shows the percentages of LSIL, HSIL, and anal cancer in the total cohort, which were 62.3%, 18%, and 1.7%, respectively, as shown.
  4. The title relates to ORAL but the article is on GENITAL and ORAL. Although we studied oral, anal, and genital mucosae, we would prefer to refer to oropharyngeal mucosa to avoid the title being too long.
